# The Influence of Injection Temperature and Pressure on Pattern Wax Fluidity

**Viacheslav E. Bazhenov** [1],*, **Andrey V. Sannikov** [1], **Elena P. Kovyshkina** [1], **Andrey V. Koltygin** [1], **Andrey I. Bazlov** [2], **Vladimir D. Belov** [1] and **Dmitry N. Dmitriev** [3]

1   Casting Department, National University of Science and Technology "MISiS", Leninskiy Pr. 4, 119049 Moscow, Russia; sannikov@ic-ltm.ru (A.V.S.); lena.kovyshkina@yandex.ru (E.P.K.); misistlp@mail.ru (A.V.K.); vdbelov@mail.ru (V.D.B.)
2   Laboratory of Advanced Green Materials, National University of Science and Technology "MISiS", Leninskiy Pr. 4, 119049 Moscow, Russia; bazlov@misis.ru
3   Public Joint Stock Company UEC "Kuznetsov", Zavodskoe Shosse 29, 443009 Samara, Russia; dn.dmitriev@uec-kuznetsov.ru
*   Correspondence: v.e.bagenov@gmail.com; Tel.: +7-(905)-553-55-64

**Abstract:** In the investment casting process, the pattern made of wax is obtained in a die for further formation of a shell mold. The problem of die-filling by pattern wax is significant because it influences the quality of the final casting. This work investigates three commercial pattern waxes' fluidity with a newly developed injection fluidity test. It was shown that the fluidity of waxes increased with increasing injection temperature and pressure, and the simultaneous increase in temperature and pressure gives a much more significant enhancement of fluidity than an increase in temperature or pressure separately. The rheological behavior of the waxes was also investigated at different temperatures using a rotational viscosimeter, and temperature dependences of waxes' dynamic viscosity were determined. It was shown that wax viscosity is increased more than ten times with decreasing temperature from 90 to 60 °C. A good correlation between wax fluidity and its viscosity is observed, which is different from metallic alloys, where the solidification behavior is more critical. The difference in wax flow behavior in comparison with metallic melts is associated with the difference in dynamic viscosity, which for investigated waxes and metallic melts is 3000–27,000 mPa·s and 0.5–6.5 mPa·s, respectively. The difference in investigated filled waxes' fluidity is observed, which can be associated with the type and amount of filler. The twice-increasing fraction of cross-linked polystyrene decreases fluidity twice. At the same time, terephthalic acid has a minor influence on wax fluidity.

**Keywords:** fluidity test; investment casting; pattern wax; viscosity; wax injection

## 1. Introduction

The investment casting process is a widespread casting technique in which a pattern is formed by injecting molten wax into a die. This pattern is further used for producing ceramic shell molds and is withdrawn by dewaxing. The advantages of investment casting are high dimensional accuracy and perfect surface quality of produced castings [1–5]. The wax pattern injection parameters influence linear shrinkage and surface finish of the wax pattern and casting [6,7].

A wax injection may be in a paste state (semi-solid) or liquid state. Paste injection allowed us to obtain patterns at low temperatures. It is preferable due to lower shrinkage and for improving the strength of the wax pattern, but it requires much higher pressures. At the same time, the injection must provide the complete filling of the die cavity. Thus, the liquid injection into the dies can be used where the pressure cannot be so high and favorable, for example, for large patterns [4,6,7]. The high injection temperature resulted in more wax pattern linear and volumetric shrinkage, increased temperature gradients,

evaporation of volatile content, and some loss of material from the wax blend, increasing the shrinkage of the wax blend [8].

The problem of die-filling by pattern wax is significant but mostly ignored, and generally, trial-and-error methods are used to adjust injection parameters [9]. However, in [2], simulations have been carried out to determine the filling of the die. The transparent die was used to visualize the wax flow for simulation results verification. The simulations allow authors to predict filling problems (short shot, formation of weld lines, etc.) with Moldex3D software [1]. In our work, the fluidity of liquid wax was under investigation, which can influence misrun and welding line formation in the thin sections of wax patterns [10,11].

The wax should have a high fluidity to fill the thinnest sections of the die [5,12]. Investment pattern waxes are blends of many compounds comprising natural or synthetic wax, natural or synthetic resin, solid organic fillers, and even water [5]. These additions are combined to achieve the best proportion of cohesiveness, surface finish, fluidity, strength, and dimensional accuracy [13,14]. As for wax fluidity, the addition of soybeans, gum arabic, silicone, etc., can increase it or, in the case of pectin, agar, polyvinylpyrrolidone-40 (PVP-40), etc., decrease it, or do not influence it [12,15,16]. Thus, the wax composition is critical for its fluidity.

The rheological behavior of the waxes is complicated and different in the liquid, paste, and solid states [17]. At high temperatures well above the melting points, all constituents of hydrocarbon waxes are Newtonian fluids with relatively low viscosity [9]. In the solid and paste states, the waxes exhibit the behavior of an elastic-viscoplastic material [14]. The waxes' non-Newtonian behavior can also be observed at high shear rates and low temperatures [18,19].

The information about the fluidity test for pattern waxes' fluidity determined in the literature was insufficient. Because of that, a new fluidity test probe was developed. Many fluidity probes for metallic alloys exist, but most are designed for gravity filling. The idea of the wax fluidity test configuration was taken from the work of Watanabe et al., where the probe with the close configuration was used to determine the titanium alloy's fluidity [20]. The developed probe must provide the wax's calm flow in the die channels due to the pins barrier effect that hinders wax velocity.

The objectives of this work were (i) to determine the fluidity of three commercial pattern waxes using the wax fluidity test; (ii) to investigate the viscosity, enthalpy of transition, and transition temperatures of waxes; and (iii) to determine the relationship between wax fluidity and mentioned wax properties.

## 2. Materials and Methods

Three brands of filled wax, namely RG20, S1235, and S1135, with different quantities and types of additives, were selected for this study. All waxes were supplied from Technopark (Russia). The wax properties provided by the supplier are presented in Table 1.

**Table 1.** Wax properties that the supplier provides.

| Wax | RG20 | S1235 | S1135 |
|---|---|---|---|
| Additives | Cross-linked polystyrene particles | Cross-linked polystyrene particles | Cross-linked polystyrene particles/terephthalic acid |
| Additives content (%) | 18 | 35 | 22/13 |
| Drop melt point (°C) | 71 | 77 | 75 |
| Viscosity at 100 °C (mPa·s) | 310 | 400 | 350 |
| Linear shrinkage (%) | 0.65 | 0.65 | 0.7 |
| Ash-content (%) | 0.02 | 0.04 | 0.03 |

The die for fluidity determination was developed and machined from two plates of 5251 aluminum alloy. Die configuration and dimensions are presented in Figure 1. The die has two halves with two locating pins on one half (not shown in the figure). The die cavity that formed a fluidity probe is located in one half. The probe is a 3 mm thick plate $80 \times 83$ mm$^2$ with 90 perforations 5 mm in diameter. As shown in Figure 1, the wax entered through the gate with a cross-section of 9.5 mm$^2$ and spread via three narrow passages with the $3 \times 3$ mm$^2$ area to fill the fluidity pattern. The die cavity perimeter has narrow flat channels for air escape.

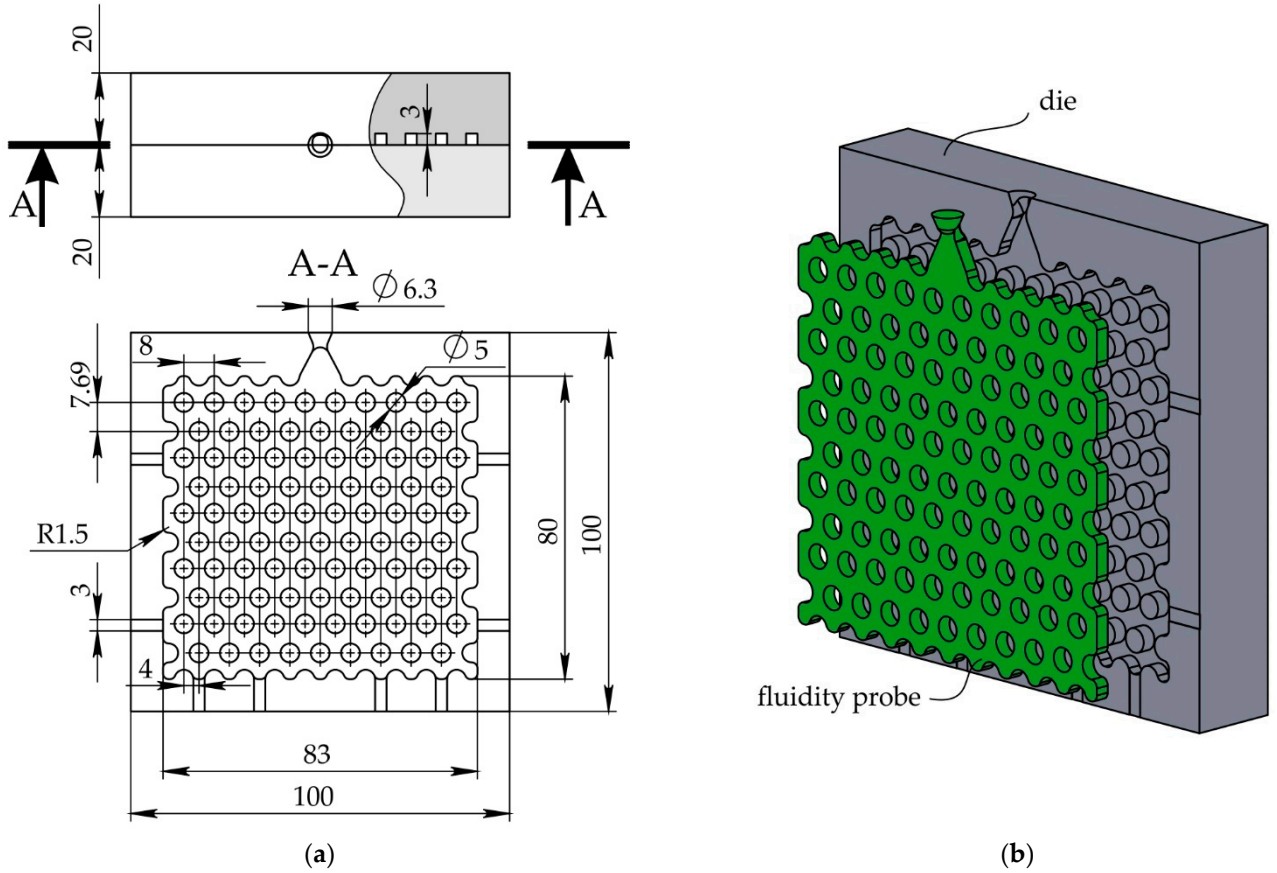

**Figure 1.** The die used for fluidity test: (**a**) top view and cross-section and (**b**) probe and die in 3D.

Injection of the die was carried out on a digital vacuum wax injector DING XIN D-VWI-1 (China), and fluidity probe patterns at different settings of injection parameters were produced. The injector can control the wax camera and nozzle temperature, which was set the same. Before the experiment, the wax was melted in the injector camera and held at 90 °C for 2 h [21]. The die was at room temperature before injecting the wax and was lubricated with oil. The molten wax was injected into the die at an injection pressure of 49, 98, 147, and 196 kPa and injection temperature from 65 to 90 °C with a 5° step. The lower injection temperatures were not analyzed because when the injection temperature <60 °C, the pressure must be higher than 1 MPa [22]. The duration of wax injection was set to 2 s. The temperature variations during injection are not significant in the bulk of the wax due to the low thermal diffusivity of the wax [10,11]. After the injection, the die was opened, and the fluidity test pattern was photographed using Canon 6D camera and removed from the die. In total, 4–5 injections were carried out for each pressure and temperature, and the die was cooled with water. After the change of injection temperature for the next series of experiments, the wax in the injector camera was held for 2 h for temperature flattening.

The fluidity was calculated as the area of the fluidity probe filled with wax divided by the area of the completely filled probe expressed in percentage. The filled and total area of

the probe was determined on photographs with the help of the ImageJ software (version 1.52a, National Institutes of Health, Bethesda, MD, USA).

The rheological behavior of the waxes was investigated using Brookfield DV2TLV rotational viscosimeter (USA). The experimental investigation of the dynamic viscosity has been performed using LV-4 (#64) spindle at a rotation frequency range of 0.017–3.333 Hz (1–200 rev·min$^{-1}$) and temperatures in the range of 60–90 °C with a 5 °C step. Before viscosity measurement, the wax was melted in the glass installed in the UT-4313 water bath (ULAB, Shanghai, China). When the wax temperature rose to 90 °C, the viscosimeter spindle was handled to wax, and its rotation started. The spindle stirred the wax for two hours to equalize the temperature in the wax–spindle system, after which measurements were taken. The measuring cup was covered with plastic wrap having a spindle access opening to prevent cooling of the surface layer of the molten wax. The duration of the one viscosity measurement was 2 min. The spindle rotational frequency was changed from the maximal value (3.333 Hz) to the minimal value (0.017 Hz), and then the rotational frequency was increased for the maximal value again (3.333 Hz). To measure the temperature of the wax, a thermocouple built into the viscometer was used, which provides an accuracy of $\pm 1$ °C in the temperature range under study. When the viscosity measurements for one temperature were finished, the preset temperature on the water bath was changed, and the spindle rotation started and lasted for 2 h for temperature flattening. Then, the next series of experiments began.

The waxes' transition temperatures and transition enthalpies were measured using a Linseis Chip DSC 100 differential scanning calorimetry (DSC) instrument with a heating and cooling rate of 10 °C·min$^{-1}$. The 10–20 mg wax samples were heated to 120 °C in the Al pans under an Ar gas flow and cooled to room temperature. The literature shows that the wax blend remelting cycles influenced shrinkage, but the melting and solidification temperatures change insignificant [21]. However, this work observed that the curve differs from the first heating run for the second and other runs. Thus, the transition temperatures and enthalpies were determined from the second heating and cooling runs.

## 3. Results

Figure 2 shows the typical fluidity probes of RG20, S1235, and S1135 waxes after injection. It can be seen that the change in injection pressure and temperature have a beneficial influence on the fluidity of waxes under investigation. The probe is well fitted to the chosen gap of injection pressures and temperatures due to the percent of die filling varying from close to zero to full (100%). Also, it was established that the probes obtained in the same conditions look amazingly the same. This behavior is very different from the fluidity test of metallic alloys. The possible reason for the excellent reproducibility of fluidity obtained by the developed wax fluidity test is in the calm flow of the wax in the die channels due to the barrier effect of the pins in the die that break the flow and decrease wax velocity. Also, the increasing wax viscosity with decreasing temperature positively influences test pattern reproducibility. Following the obtained results, the shape of the filling front is a semicircle in most cases. But when the die was filled for more than 50%, and the side walls were filled, the shape of the front was flat (not provided in Figure 2).

The influence of injection temperature and pressure on the fluidity of RG20, S1235, and S1135 waxes is shown in Figure 3. For all investigated waxes, the dependence of temperature vs. fluidity is linear, with the coefficient of determination at 0.98–0.99. Similarly, the influence of injection pressure on waxes' fluidity is also linear (not shown), with one exception for the lowest injection temperature of 65 °C.

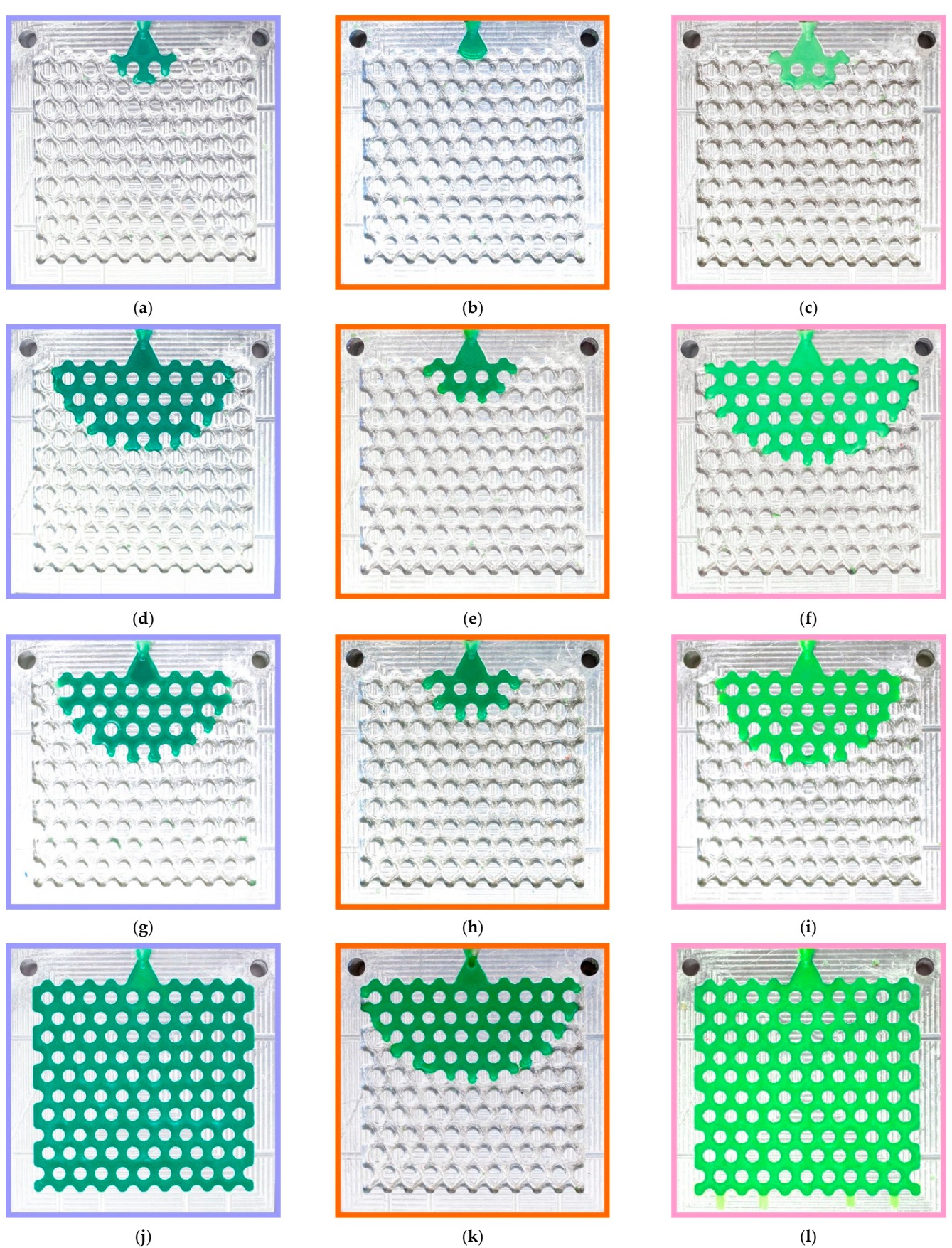

**Figure 2.** Typical fluidity probes of (**a**,**d**,**g**,**j**) RG20, (**b**,**e**,**h**,**k**) S1235, and (**c**,**f**,**i**,**l**) S1135 waxes at (**a**–**f**) 70 and (**g**–**l**) 90 °C injection temperatures and (**a**–**c**,**g**–**i**) 49 and (**d**–**f**,**j**–**l**) 196 kPa injection pressures.

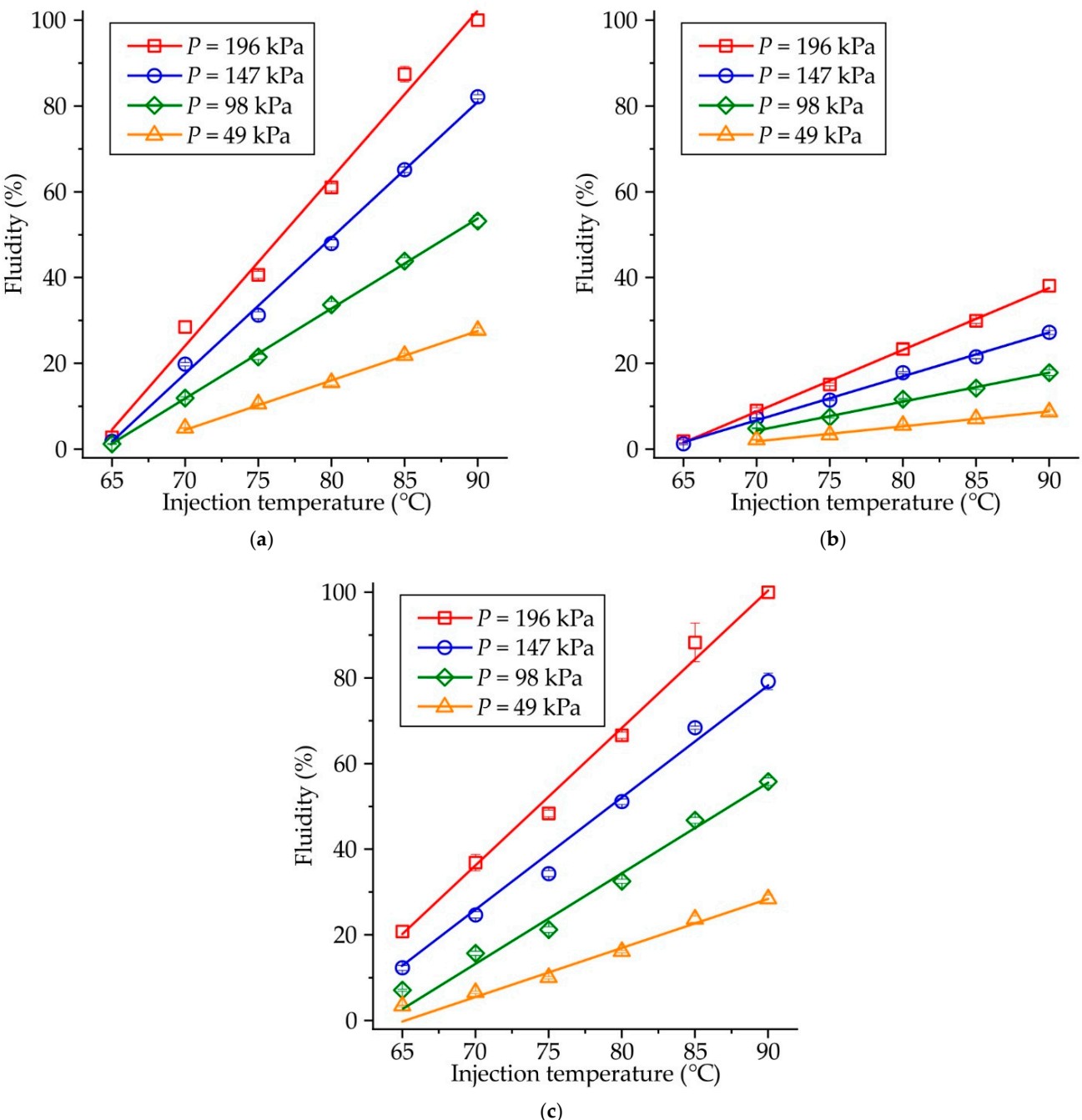

**Figure 3.** Fluidity of (**a**) RG20, (**b**) S1235, and (**c**) S1135 waxes at various injection temperatures and pressures.

Figure 4 provides the contour plots showing the influence of injection pressure and temperature on the fluidity of RG20, S1235, and S1135 waxes. Experimental data were fitted to surface equations using the least square method, and Equations (1)–(3) for RG20, S1235, and S1135 waxes, respectively, were obtained. Following provided graphs and equations for all waxes, increasing injection pressure and temperature increases fluidity. For provided equations, the coefficient of determination was 0.99, which is higher in comparison with the determination coefficient when obtained for data planar surface fitting (~0.9). That means that a simultaneous increase in temperature and pressure gives a much more significant enhancement of fluidity than an increase in temperature or pressure

separately. Equations (1)–(3) are valid for the injection temperature range of 60–90 °C and injection pressure range of 49–196 kPa.

$$\text{Fluidity (\%)} = 0.23 \cdot T - 1.21 \cdot P + 0.019 \cdot T \cdot P - 17.66, \tag{1}$$

$$\text{Fluidity (\%)} = -0.065 \cdot T - 0.5 \cdot P + 0.008 \cdot T \cdot P + 4.63, \tag{2}$$

$$\text{Fluidity (\%)} = 0.41 \cdot T - 0.86 \cdot P + 0.015 \cdot T \cdot P - 32.23, \tag{3}$$

The injection temperature is $T$, and the injection pressure is $P$ here.

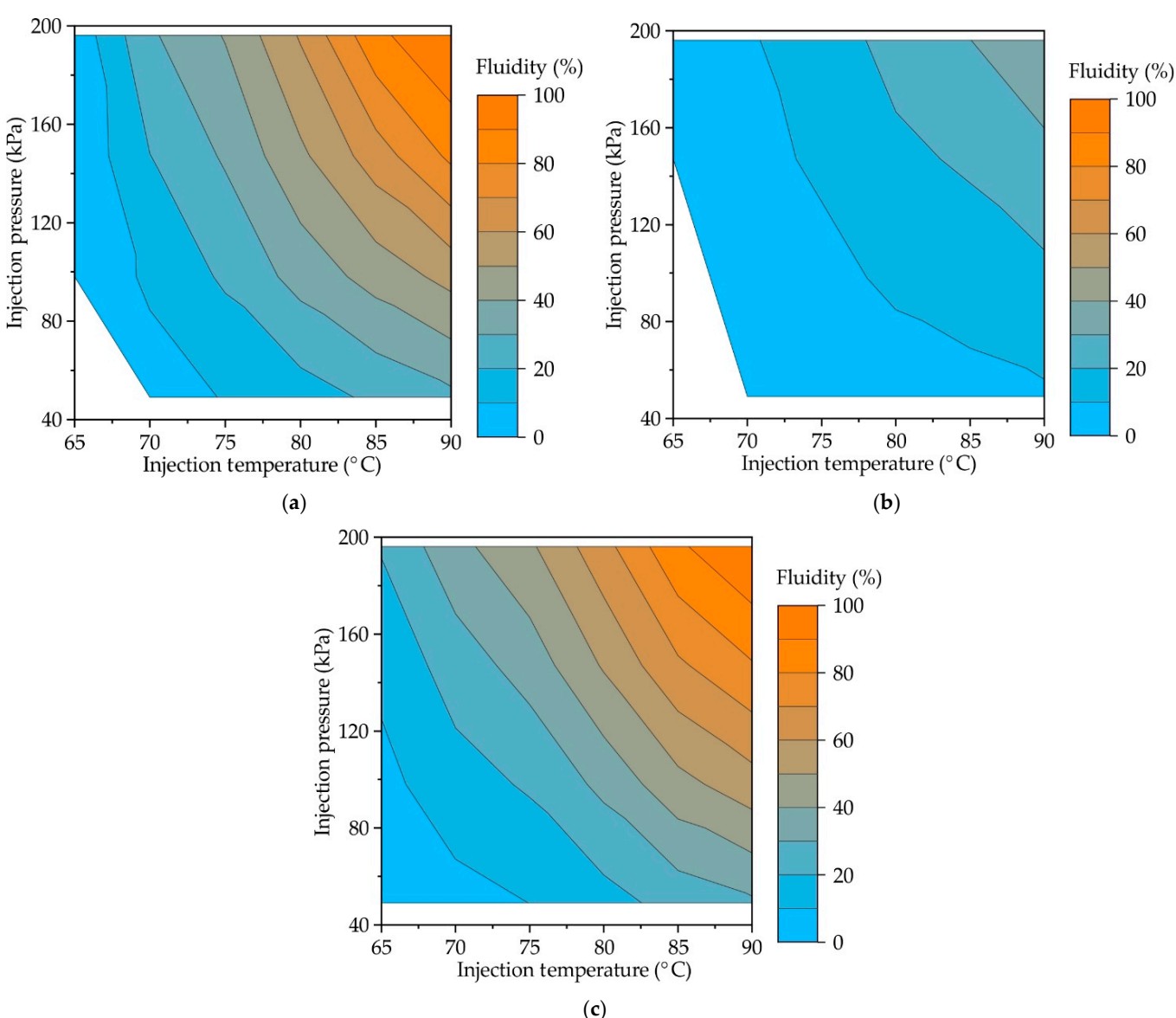

**Figure 4.** Contour plot showing the influence of injection temperatures and pressures on the fluidity of (**a**) RG20, (**b**) S1235, and (**c**) S1135 waxes.

The fluidity at the same injection conditions is not the same for waxes under investigation due to their different composition. Figure 5 shows the influence of injection temperature on the fluidity of RG20, S1235, and S1135 waxes at constant injection pressure. When the injection pressure is low (*P* = 49 kPa, Figure 5a), the fluidity of RG20 and S1135 waxes is nearly the same, but the fluidity of S1235 wax is almost three times lower. For example, at an injection temperature of 90 °C, the fluidity of RG20, S1235, and S1135 waxes

are 27.7, 8.7, and 28.4%. However, with the increasing injection pressure (Figure 5b–d), the difference between the fluidity of RG20 and S1135 waxes is observed at low injection temperatures (60–70 °C). The fluidity of mentioned waxes is near the same for high injection temperatures (80–90 °C). Interestingly, at an injection temperature of 65 °C and injection pressure of 196 kPa, a very low fluidity (<3%) is observed for RG20 and S1235 waxes. Conversely, the S1135 wax shows more than seven times greater fluidity of 20.8%. The lowest fluidity in all conditions is observed for S1235 wax. For instance, at injection temperature and pressure of 90 °C and 196 kPa, respectively, the fluidity of RG20 and S1135 waxes is 100%, but the fluidity of S1235 wax is only 38.1% (Figure 5d).

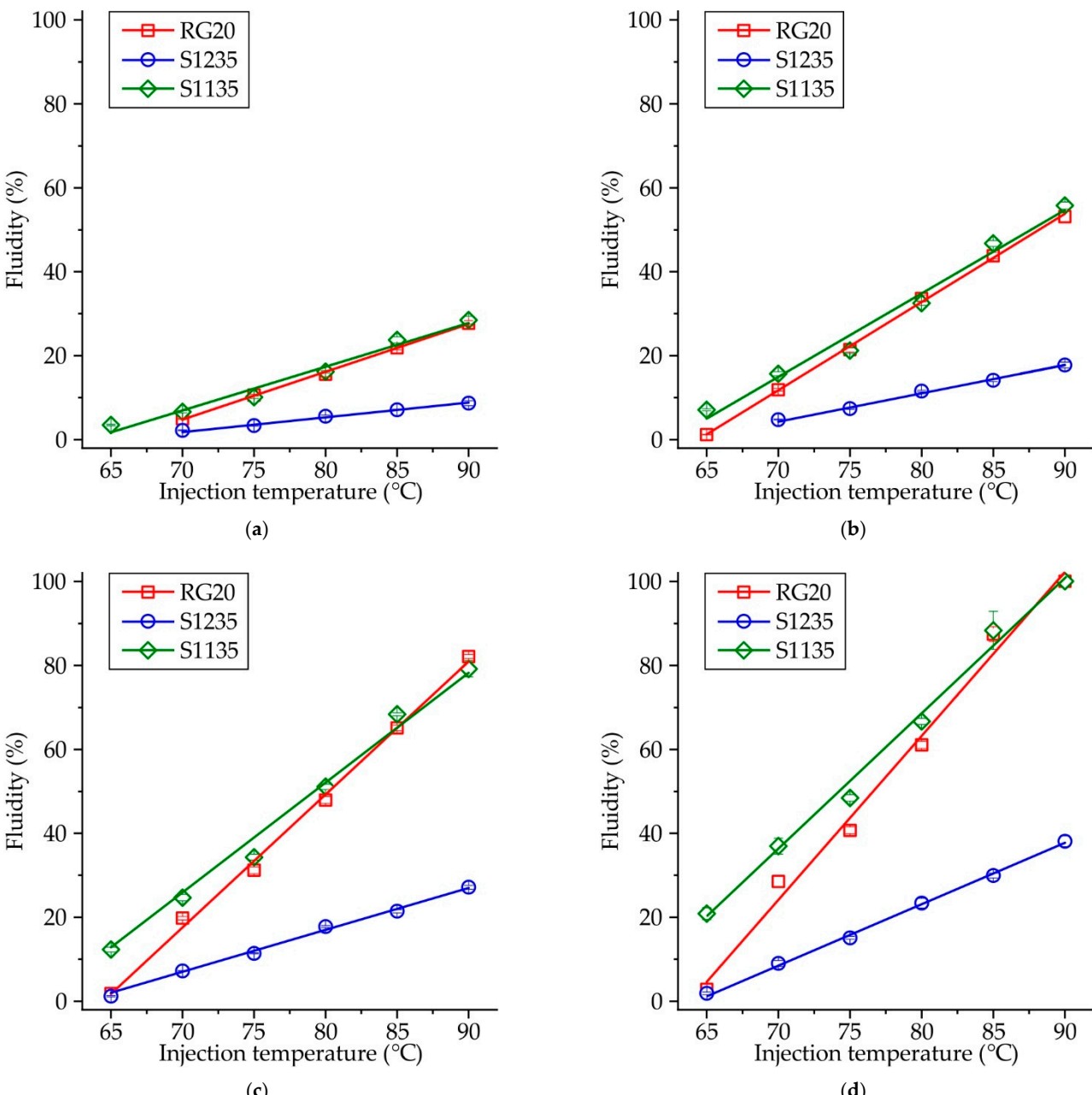

**Figure 5.** Fluidity of RG20, S1235, and S1135 waxes depending on injection temperatures at pressures (**a**) *P* = 49 kPa, (**b**) *P* = 98 kPa, (**c**) *P* = 147 kPa, and (**d**) *P* = 196 kPa.

Figure 6a–c shows the influence of spindle rotational frequency on the viscosity of RG20, S1235, and S1135 waxes at various temperatures. When the temperature of the wax is high, the dynamic viscosity is almost constant and does not depend on the

rotational frequency of the viscosimeter spindle (shear rate). At mentioned temperatures, the waxes demonstrate the behavior of a Newtonian fluid. When the wax temperature is decreased, the wax rheological behavior changes to non-Newtonian, and at low spindle rotation frequency, the viscosity is higher than at high rotation frequency. It agrees with the literature data [9,14,17–19]. The transition point, when the rheological behavior change from Newtonian to non-Newtonian ('congealing point'), is close to 70 °C for RG20 wax, in the range of 75–80 °C for S1235 wax, and 70–75 °C for S1135 wax. These points correspond well to the values of the drop melt point of waxes provided by the supplier. Following Table 1, the drop melt points are 71, 77, and 75 °C for RG20, S1235, and S1135 waxes, respectively. Previous experiments show that, unlike other homogeneous chemical compounds, wax does not melt immediately on heating but passes through several intermediate states: solid plastic, semi-plastic, semi-liquid, and liquid. The wax exhibits Newtonian behavior when fully liquid and non-Newtonian behavior when semi-liquid [18,19].

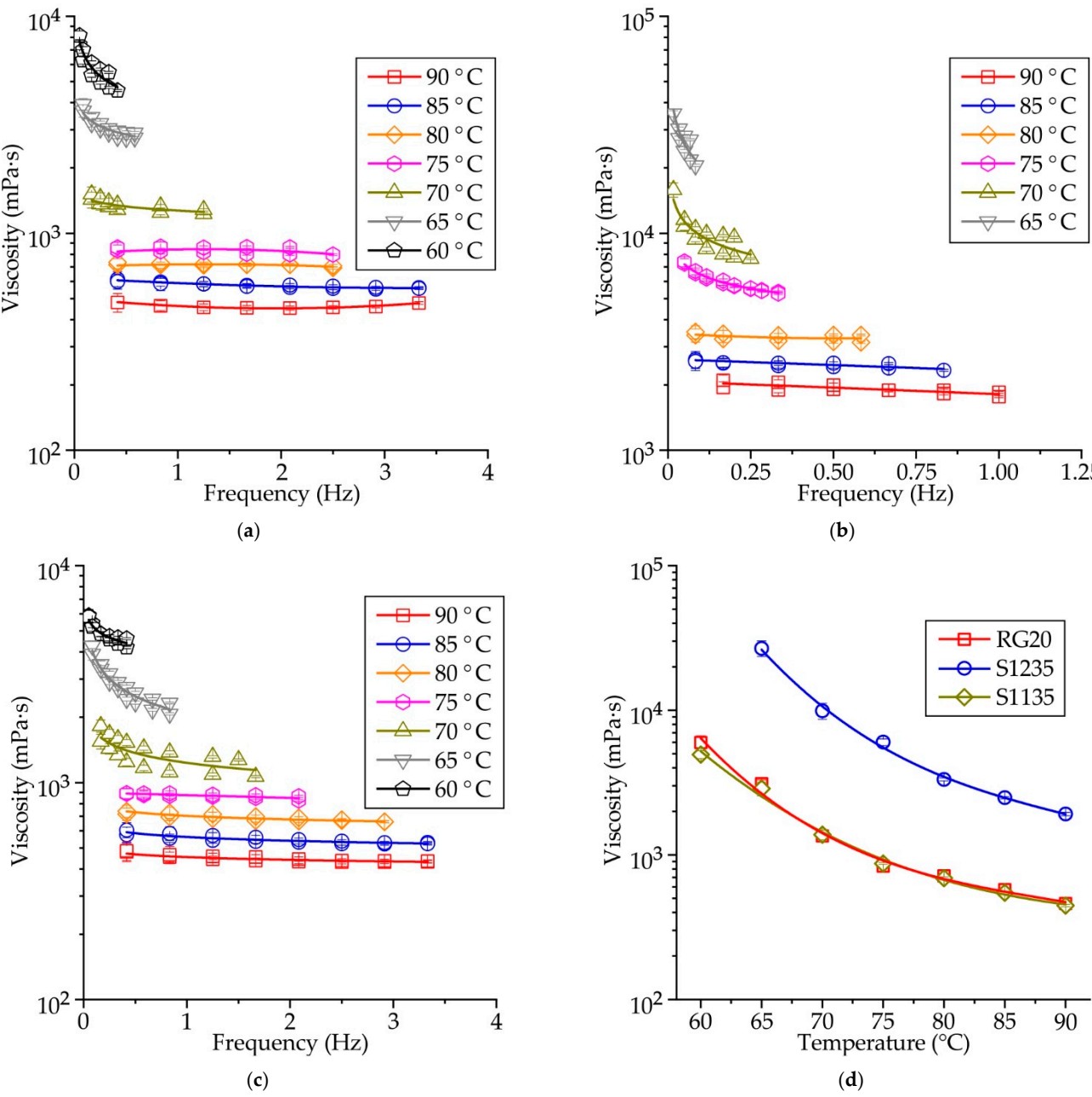

**Figure 6.** The influence of rotational frequency on the viscosity of (**a**) RG20, (**b**) S1235, and (**c**) S1135 waxes at various temperatures and (**d**) mean viscosity at various temperatures of waxes.

The mean values of dynamic viscosity calculated for all spindle rotation frequencies for RG20, S1235, and S1135 waxes are shown in Figure 6d. It can be seen that the wax viscosity increases with decreasing temperature. For example, the viscosity of RG20 wax at temperatures of 90, 80, 70, and 60 °C are 462, 715, 1349, and 5973 mPa·s, respectively. Thus, when the temperature decreased from 90 to 60 °C, the viscosity of RG20 increased 13 times. For S1235 and S1135 waxes in mentioned temperature range, the viscosity increased by 11–14 times.

When the waxes are heated to a temperature of 90 °C, the viscosity of the RG20, S1235, and S1135 waxes are 462, 1917, and 447 mPa·s. Thus, the values of viscosity provided in Table 1 and obtained by the supplier are quite probable, with the one exception for S1235 wax, which has a very high viscosity. The comparison of waxes' viscosities allows us to say that the viscosity of the RG20 and S1135 waxes are very close at high temperatures. Still, when the temperature is decreased, the difference is meaningful. For example, at 60 °C, RG20 and S1135 waxes viscosity is 5973 and 4933 mPa·s. As for S1235 wax viscosity, it is more than four times higher than for other investigated waxes at 90 °C and nearly nine times higher when the temperature decreased to 65 °C (the viscosity at 60 °C of S1235 wax is outside the measurements range of viscometer with the used model of the spindle). As usual lower molecular weight resulting lower viscosity of wax [23]. On the other hand, the behavior of the polyethylene/wax blends depends not only on the structure and molecular weight of the waxes but also on the structure and fraction of the polyethylene [23].

Figure 7 presents the second run heating and cooling DSC curves of RG20, S1235, and S1135 waxes. The liquidus and solidus temperature of the waxes under investigation is very close and in the range of 65.8–66.6 °C and 27.5–29.1 °C, respectively. Thus, the freezing range of the waxes is near the same and close to 38 °C. The long freezing range of the investigated waxes is because the waxes are mixtures of compounds that solidify over a temperature range and contain a wide range of molecular weights [24]. The 3–4 peaks of heat release are observed when the wax is solidified. It is known that the wax is very polydisperse, and molecules of different molecular weights attempt to crystallize at different temperatures [9]. Also, the wax is a multi-component mixture, and the components of the mixture can crystallize at different temperatures. The same situation can be seen for heating curves because of the melting of crystalline fractions with different molecular weight distributions [10]. Also, the transition of one crystalline phase into another, a so-called solid–solid transition, is typical for waxes. As for transition enthalpy calculated via DSC curves, the enthalpy of solidification for RG20, S1235, and S1135 waxes was 42.9, 36.4, and 47.8 J·g$^{-1}$, respectively.

Interestingly, heat absorption is lower when melting than heat release during solidification for all waxes, which can be associated with the waxes' partially amorphous and partially crystalline behavior [24]. During cooling, some wax is amorphized and does not yield crystallization heat releases. When heated, we see the superposition of the heat absorption during the melting of the crystalline phase and heat release due to the crystallization of the amorphous phase. The maximal difference between the melting and solidification heat is observed for S1235 wax with the lowest fluidity.

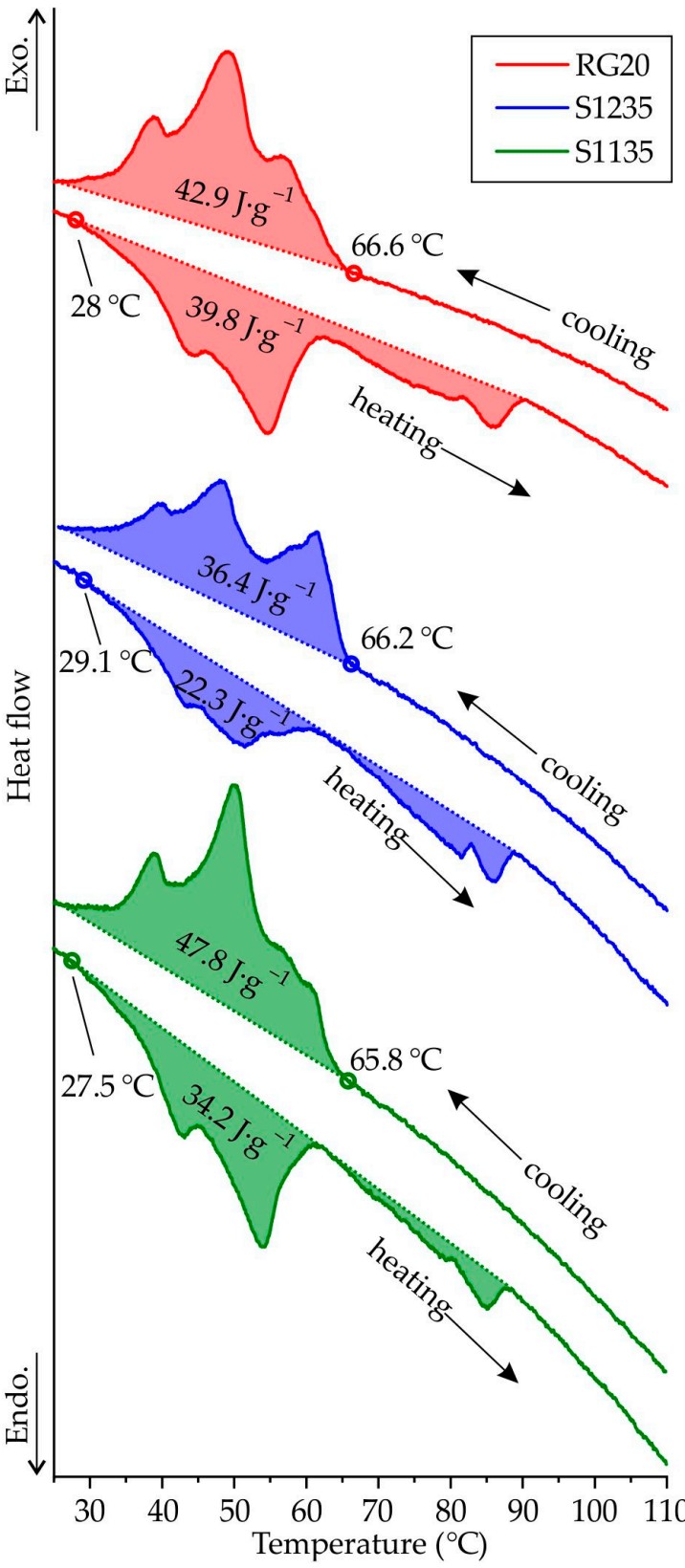

**Figure 7.** The heating and cooling DSC curves for RG20, S1235, and S1135 waxes.

## 4. Discussion

Fluidity is a complex technological property, and it depends upon many factors. For metallic melts, the variables influencing fluidity are freezing range, viscosity, and heat

of fusion [25]. Following the DCS results, the investigated waxes' liquidus temperatures and freezing range are nearly identical. Moreover, the viscosity of investigated waxes at liquidus temperature ~66 °C is 3000–27,000 mPa·s, which is very high, and the wax flow in the die cavity is seriously hindered. For comparison, the dynamic viscosity of liquid metals at liquidus temperature is 0.5–6.5 mPa·s, which is four orders of magnitude lower [26]. At the same time, for RG20, S1235, and S1135 waxes, when the temperature decreases from 90 to 65 °C, the viscosity increases by 11–14 times, but the viscosity of metallic melts little changes with temperature [26]. The same wax rheological behavior can be seen in other works [18,19]. It is known that the solid particles of additives cause the filled waxes to behave as non-Newtonian fluids even when well above the melting point for the wax [9]. The presence of both fillers and resins in pattern waxes makes their rheological behavior decidedly non-Newtonian and makes traditional viscometry less applicable [9]. The maximal and minimal heat released on solidification is observed for S1135 and S1235 waxes, which show maximal and minimal fluidity, respectively. However, the fluidities of RG20 and S1135 waxes are very close and significantly different from that of S1235 wax. Also, the heat released during wax solidification (36.4–47.8 J·g$^{-1}$) is nearly ten times lower than for metallic alloys. For instance, the melting enthalpy of the A356 (Al-7Si) and Al-12Si alloys were ~410 and 530 J·g$^{-1}$, respectively [27–30]. It means that heat released during wax solidification is not strongly proportional to fluidity, and other properties of wax significantly affect its fluidity. Further, the results of the numerical simulations indicate that due to its low thermal diffusivity and thermal conductivity (0.2–0.4 W·m$^{-1}$·K$^{-1}$ [18,22,31–34]), when the wax is in the die, a thin layer of solid wax forms on the surface of the pattern, the wax cools very slowly [31].

The relationship between wax viscosity and fluidity was known and found previously [12,15]. Following obtained results, the viscosity for RG20 and S1135 over the investigated temperature range is nearly the same. At the same time, the fluidity of these waxes is also very close, especially at high injection temperatures. The only deviation from this behavior is observed at low injection temperature (65 °C) and high injection pressure (196 kPa). In this case, the fluidity of S1135 wax is significantly higher than that of RG20. Although, as mentioned earlier, at low temperatures, the difference in dynamic viscosity of these waxes is more significant than at high temperatures, and viscosity is lower for S1135 wax, there are probably other reasons for the fluidity differences for these waxes. This result may be due to the coupled effect of lower viscosity and higher heat released during S1135 wax solidification. Also, the rheological behavior of wax at temperatures below 65 °C was not studied in this work. Perhaps already in the semi-solid (paste) state, the S1135 wax has a higher ability to flow, and the difference in viscosity between S1135 and RG20 waxes increases even more.

Figure 8 presents the influence of investigated waxes' viscosity on their fluidity. Each point corresponds to fluidity and viscosity values at the same temperature. This approach is not entirely correct since the fluidity is determined by the behavior of the wax over a wide temperature range from the injection temperature to low temperatures. However, this allows us to see a reasonably good correlation between the waxes' viscosity and fluidity. For example, when the injection pressure is 49 kPa, the curves show decreasing fluidity with increasing viscosity which is the same for RG20 and S1135 waxes. At the same time, the viscosity of S1235 wax is higher and fluidity is lower, and the curve for S1235 wax looks like a continuation of the curves for RG20 and S1135 waxes. The same behavior is observed for higher injection pressure of 196 kPa, and the curve for S1235 wax continues the S1135 curve. The increasing injection temperature leads to increasing fluidity due to the high heat content of the wax when the temperature is increased, and the broader time in that wax has a low viscosity.

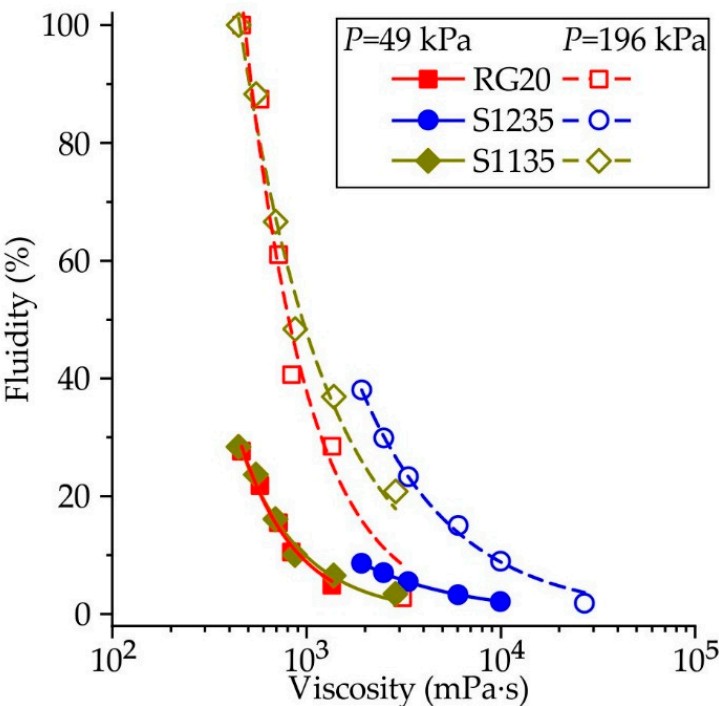

**Figure 8.** The influence of RG20, S1235, and S1135 waxes viscosity on its fluidity at different injection pressures. Each fluidity point corresponds to the viscosity at the same temperature.

Under the results of this work for all investigated waxes, increasing injection pressure leads to increased fluidity. It was previously discussed that viscosity, which affects wax fluidity, depends on temperature and shear rate [17]. Following the literature, the change of shear rate from 0.01 to 100 $s^{-1}$ leads to decreasing wax viscosity up to $10^4$ times [34]. With injection pressure increasing, the shear rate of wax increases, and its viscosity is decreased. As shown in Figure 6, increasing the shear rate for investigated waxes leads to decreasing viscosity, but only at low temperatures, where waxes show non-Newtonian behavior. Also, with high injection pressure and higher velocity, the wax can cool less due to lower contact time between the wax and the die.

The fluidity of RG20 and S1135 waxes is near the same due to close dynamic viscosity values. Following the information the wax manufacturer provided, the additions in waxes are different. Powdered, inert fillers are added to investment casting waxes for various reasons, including improvements in structural stability, mold release, strength, dimensional stability, compatibility with mold materials, and economy [9]. Compared with RG20 wax in the S1135 wax, the addition of terephthalic acid $C_6H_4(CO_2H)_2$ is presented. At the same time, the content of cross-linked polystyrene $(C_8H_8)_n$ for RG20 and S1135 is very close (18 and 22%, respectively). That means that terephthalic acid has a low influence on wax viscosity and, thus, on fluidity, as shown previously. At the same time, increasing cross-linked polystyrene content in wax from 18 to 35% leads to decreasing fluidity more than two times, as shown for RG20 and S1235 waxes. The reason for that is increasing the viscosity of S1235 wax up to 9 times compared with RG20 wax.

Furthermore, increasing polystyrene fraction increased melting drop points from 71 to 77 °C for RG20 and S1235 waxes, respectively. However, adding terephthalic acid also changes the drop point from 71 to 75 °C for RG20 and S1135 waxes, respectively. Thus, no strong correlation between the fluidity of the wax and its drop melting point is found.

## 5. Conclusions

The fluidity of three commercial waxes (RG20, S1235, and S1135) was determined with the help of a newly developed fluidity test. The results show the linear dependence for the influence of injection pressure and temperature on fluidity. It was established that

with decreasing temperature from 90 to 60 °C, the wax viscosity is increased more than ten times, which is a primary factor in decreasing wax fluidity.

Increasing the quantity of polymer additives, like cross-linked polystyrene, from 18 to 35% increases wax viscosity by four times and decreases fluidity by half. The transition temperatures of all investigated waxes obtained via DSC are nearly the same, and the enthalpy of solidification is not correlating with fluidity.

Overall, the main factor that depends on wax fluidity is its viscosity which changes significantly in the range of temperatures and shear rates under investigation. Thus, the waxes show different behavior than metallic alloys, for which the main parameters influencing fluidity are freezing range and heat released during solidification.

Further work will focus on the simulation of misrun prediction when the wax pattern forms in the die. The problem with this simulation is that engineering properties cannot be used to determine wax flow in the pattern die, and much more properties are needed [17,31,32]. This work determined the viscosity and heat released during wax solidification.

**Author Contributions:** Conceptualization, V.E.B.; methodology, A.V.S.; software, V.E.B.; validation, A.V.K., A.I.B. and A.V.S.; formal analysis, E.P.K.; investigation, V.E.B., A.V.S., E.P.K. and A.I.B.; resources, D.N.D.; data curation, A.V.K.; writing—original draft preparation, V.E.B.; writing—review and editing, V.E.B., A.V.K. and A.I.B.; visualization, A.V.S.; supervision, A.V.K. and V.D.B.; project administration, A.V.K., V.D.B. and D.N.D.; funding acquisition, V.D.B. and D.N.D. All authors have read and agreed to the published version of the manuscript.

**Funding:** This research received financial support from the Ministry of Science and Higher Education in the Russian Federation (Agreement No. 075-11-2022-023 from 6 April 2022) under the program "Scientific and technological development of the Russian Federation" according to governmental decree N 218 dated 9 April 2010.

**Data Availability Statement:** The data presented in this study are available on request from the corresponding author.

**Acknowledgments:** Sincere gratitude is given to Anna A. Nikitina from NUST MISIS for help processing pictures of fluidity test probes.

**Conflicts of Interest:** The authors declare no conflict of interest.

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
