# Peer review of "The Influence of Injection Temperature and Pressure on Pattern Wax Fluidity"

_jmmp, doi:10.3390/jmmp7040141_

Round 1
Reviewer 1 Report
Review of the work “Influence of Injection Temperature and Pressure on Pattern Wax Fluidity.”
This work is focused on determining the fluidity of three commercial waxes used for elaborating patterns used in the investment casting process using a prototype mold. Also, the authors determine, for the waxes under study, the viscosity, the enthalpy of fusion and solidification, and the temperatures of these transitions with the ultimate goal of finding an explanation of the fluidity behavior based on these properties. The authors study, for the three mixtures of commercial waxes under study, the effect of the change in temperature and injection pressure on the fluidity during the filling of a prototype mold specially designed to study fluidity, previously reported in the literature. Using differential scanning calorimetry (DSC), they determine the liquidus and solidus temperatures of the waxes under study and the corresponding enthalpies of fusion and solidification. They also determine the viscosities of these mixtures at different temperatures involving the waxes being in the liquid or semi-solid state and at different strain rates. The results obtained by the authors indicate that an increase in temperature and injection pressure increase the fluidity shown by the three waxes and that two of the three waxes analyzed present relatively similar fluidity behaviors. The authors use the viscosity measurements of the waxes at different temperatures and strain rates, as well as the determinations of the transition temperatures and enthalpies, to search for a connection between these properties and the observed changes in fluidity. This is excellent work, which is relevant for obtaining patterns and then parts through the investment casting process and specifically in selecting the wax mixtures used and the injection conditions (T, P, and injection speed) used to make patterns. It should be noted that the foregoing is a critical aspect of these processes and that open information is scarce. For all of the above, I consider that this work should be published in JMMP after addressing the following minor corrections:
C1: Please include one paragraph explaining why the selected mold prototype was chosen, among others, alternative mold designs for fluidity determination found in the literature.
C2: The purpose of obtaining the sample´s photographs is not clearly explained. These photos of the solidified probes are used to determine the filling percentage of the mold? Please include one paragraph explaining how the fluidity percentage is obtained in more detail.
C3: There are errors in equation (2) on page 6; when substituting the extreme values of temperature, 90 C, and injection pressure,196 MPa, the equation gives an erroneous value; please correct this.
C4: Figure 3 shows that the S1135 wax at 65oC can show a fluidity greater than zero and that it increases at this low temperature as the injection pressure increases, unlike the other two waxes; In Figure 5a), which corresponds to the most downward injection pressure, only data of almost zero fluidity are shown for 1135. The analysis of Figure 5 a-d indicates that wax 1135 is the most sensitive to improve its fluidity as the injection pressure increases at low temperatures; propose an explanation for this behavior.
C5: Correct the following typo on page 4, second paragraph:
Say: The wax samples of I am running a few minutes late; my previous meeting is running over. 10-20 mg was heated to 120 °C in the Al pans under an Ar gas flow and cooled to room temperature.
Must say: The wax sample of 10-20 mg was heated to 120 °C in the Al pans under an Ar gas flow and cooled to room temperature.
Author Response
Review of the work “Influence of Injection Temperature and Pressure on Pattern Wax Fluidity.”
This work is focused on determining the fluidity of three commercial waxes used for elaborating patterns used in the investment casting process using a prototype mold. Also, the authors determine, for the waxes under study, the viscosity, the enthalpy of fusion and solidification, and the temperatures of these transitions with the ultimate goal of finding an explanation of the fluidity behavior based on these properties. The authors study, for the three mixtures of commercial waxes under study, the effect of the change in temperature and injection pressure on the fluidity during the filling of a prototype mold specially designed to study fluidity, previously reported in the literature. Using differential scanning calorimetry (DSC), they determine the liquidus and solidus temperatures of the waxes under study and the corresponding enthalpies of fusion and solidification. They also determine the viscosities of these mixtures at different temperatures involving the waxes being in the liquid or semi-solid state and at different strain rates. The results obtained by the authors indicate that an increase in temperature and injection pressure increase the fluidity shown by the three waxes and that two of the three waxes analyzed present relatively similar fluidity behaviors. The authors use the viscosity measurements of the waxes at different temperatures and strain rates, as well as the determinations of the transition temperatures and enthalpies, to search for a connection between these properties and the observed changes in fluidity. This is excellent work, which is relevant for obtaining patterns and then parts through the investment casting process and specifically in selecting the wax mixtures used and the injection conditions (T, P, and injection speed) used to make patterns. It should be noted that the foregoing is a critical aspect of these processes and that open information is scarce. For all of the above, I consider that this work should be published in JMMP after addressing the following minor corrections:
C1: Please include one paragraph explaining why the selected mold prototype was chosen, among others, alternative mold designs for fluidity determination found in the literature.
Answer: This paragraph was added to the introduction part: “The information about fluidity test for pattern waxes fluidity determining was not found in the literature. Because of that new fluidity test probe was developed. Many fluidity probes for metallic alloys exist, but most are designed for gravity filling. The idea of the wax fluidity test configuration was taken from the work of Watanabe et al., where the probe with the close configuration was used for determining titanium alloys fluidity [20]. This probe provides the calm flow of the wax in the die channels due to the pins' barrier effect that hinders wax velocity.”
C2: The purpose of obtaining the sample´s photographs is not clearly explained. These photos of the solidified probes are used to determine the filling percentage of the mold? Please include one paragraph explaining how the fluidity percentage is obtained in more detail.
Answer: This part was added: “The fluidity was calculated as the area of the fluidity probe filled with wax divided by the area of the completely filled probe expressed in percentage. The filled and total area of the probe was determined on photographs with the help of the ImageJ 1.52a software (National Institutes of Health, USA).”
C3: There are errors in equation (2) on page 6; when substituting the extreme values of temperature, 90 C, and injection pressure,196 MPa, the equation gives an erroneous value; please correct this.
Answer: Thank you for finding this mistake. We fix the equation.
C4: Figure 3 shows that the S1135 wax at 65oC can show a fluidity greater than zero and that it increases at this low temperature as the injection pressure increases, unlike the other two waxes; In Figure 5a), which corresponds to the most downward injection pressure, only data of almost zero fluidity are shown for 1135. The analysis of Figure 5 a-d indicates that wax 1135 is the most sensitive to improve its fluidity as the injection pressure increases at low temperatures; propose an explanation for this behavior.
Answer: This part was added: “The only deviation from this behavior is observed at low injection temperature (65 °C) and high injection pressure (196 kPa). In this case, the fluidity of S1135 wax is significantly higher than that of RG20. Although, as mentioned earlier, at low temperatures the difference in dynamic viscosity of these waxes is more significant than at high temperatures, and viscosity is lower for S1135 wax, there are probably other reasons of the fluidity differences dor this waxes. This result may be due to the coupled effect of lower viscosity, higher heat released during S1135 wax solidification. Also, the rheological behavior of wax at temperatures below 65 °C was not studied in this work. Perhaps already in the solid-liquid (paste) state the S1135 wax has a higher ability to flow and the difference in viscosity between S1135 and RG20 waxes increases even more.”
C5: Correct the following typo on page 4, second paragraph:
Say: The wax samples of I am running a few minutes late; my previous meeting is running over. 10-20 mg was heated to 120 °C in the Al pans under an Ar gas flow and cooled to room temperature.
Must say: The wax sample of 10-20 mg was heated to 120 °C in the Al pans under an Ar gas flow and cooled to room temperature.
Answer: We correct this.
Reviewer 2 Report
I have the following comments on the article:
1. Expand the abstract by e.g. the methods used as well as the main result
2. Change Table 1 to the style according to the template
3. In part 3. Results - the beginning of the sentence Fig. Change 2 to Figure and also 2, 3, 4 ... later in the text
4. Formula 1,2,3 to separate numbers and signs, it is dense
5. p. 9, paragraph under fig. 6 - 462, 715, 1.349, and 5.973 mPa·s - change the decimal point to a period.
6. p. 10 last sentence - change J/g to J*g and keep the same style as the previous one (mPa·s) - you have it like that in the graph below
7. p. 12 at the beginning ....66°C is very high, and the wax flow in the die cavity is seriously hindered. For comparison, the viscosity of investigated waxes was 3,000-27,000 mPa s at 65°C... it should be 3,000 - 27,000,
8. add - Institutional Review Board Statement, Informed Consent Statement, Data Availability Statement, Acknowledgments even if not applicable.
Author Response
- Expand the abstract by e.g. the methods used as well as the main result
Answer: The abstract was improved: “In the investment casting process, the pattern made of wax is obtained in a die for further formation of shell mold. The problem of die-filling by pattern wax is significant because it influences the quality of the final casting. This work investigates three commercial pattern waxes fluidity with a newly developed injection fluidity test. It was shown that the fluidity of waxes increased with increasing injection temperature and pressure. The rheological behavior of the waxes was also investigated at different temperatures using a rotational viscosimeter, and temperature dependences of waxes viscosity were determined. A good correlation between wax fluidity and its viscosity is observed, which is different from metallic alloys, where the solidification behavior is more critical. The difference in investigated filled waxes fluidity is observed that can be associated with the type and amount of filler.”
- Change Table 1 to the style according to the template
Answer: We fixed the style of the table.
- In part 3. Results - the beginning of the sentence Fig. Change 2 to Figure and also 2, 3, 4 ... later in the text
Answer: We fix this.
- Formula 1,2,3 to separate numbers and signs, it is dense
Answer: We fix this.
- p. 9, paragraph under fig. 6 - 462, 715, 1.349, and 5.973 mPa·s - change the decimal point to a period.
Answer: We fix this.
- p. 10 last sentence - change J/g to J*g and keep the same style as the previous one (mPa·s) - you have it like that in the graph below
Answer: We change to the same style all units.
- p. 12 at the beginning ....66°C is very high, and the wax flow in the die cavity is seriously hindered. For comparison, the viscosity of investigated waxes was 3,000-27,000 mPa s at 65°C... it should be 3,000 - 27,000,
Answer: This part was rewritten as follows: “Moreover, the viscosity of investigated waxes at liquidus temperature ~ 66°C are 3000 - 27000 mPa·s that is very high, and the wax flow in the die cavity is seriously hindered. For comparison, the dynamic viscosity of liquid metals at liquidus temperature is in the range of 0.5 - 6.5 mPa·s, that three orders of magnitude lower [26].”
- add - Institutional Review Board Statement, Informed Consent Statement, Data Availability Statement, Acknowledgments even if not applicable.
Answer: That sections were added.
Reviewer 3 Report
Comments have been attached

Author Response
Dear Authors This article deals with issues related to the foundry industry, in particular investment castings. Although 3D printing is now more and more often used to produce models instead of wax injection, the problem of wax injection is still valid. Below are some comments to make the article even better.
- The abstract should be corrected. It should contain information about the purpose of the research undertaken and the most important results obtained. The research methods used should also be stated. The abstract should encourage the reader to read the entire article.
Answer: The abstract was improved: “In the investment casting process, the pattern made of wax is obtained in a die for further formation of shell mold. The problem of die-filling by pattern wax is significant because it influences the quality of the final casting. This work investigates three commercial pattern waxes fluidity with a newly developed injection fluidity test. It was shown that the fluidity of waxes increased with increasing injection temperature and pressure. The rheological behavior of the waxes was also investigated at different temperatures using a rotational viscosimeter, and temperature dependences of waxes viscosity were determined. A good correlation between wax fluidity and its viscosity is observed, which is different from metallic alloys, where the solidification behavior is more critical. The difference in investigated filled waxes fluidity is observed that can be associated with the type and amount of filler.”
- Table 1. - please cite the company catalog or other official documents that describe the properties listed in the table.
Answer: Unfortunately, this is impossible, as the manufacturer only sends these catalogs upon request. For some reason, they are not available on the website. In addition, these catalogs are in Russian, which is inconvenient.
- Equation (1), (2) and (3) - the coefficients in the equations are given with an accuracy of five decimal places. The accuracy of two decimal places is probably enough. If the Authors claim that the values of the coefficients should be given with an accuracy of five decimal places, please justify it in the article.
Answer: We calculated the fluidity values using equations after decreasing decimal symbols in equations, and no meaningful difference was observed. Thus, we reduced the number of decimal symbols in equations (1)-(3).
- Equations (1), (2) and (3) are not discussed. On the left side of the equation, we have the percentages and the physical units of temperature and pressure on the right side. Therefore, in order to maintain the consistency of units, the coefficients should have physical names. The coefficients should also be presented as general numbers and explain their physical meaning. An example would be the known relationship between stress and strain. These quantities are related by a coefficient defined as the coefficient of elasticity or modulus of elasticity.
Answer: Fluidity is a technological property, and it is difficult to explain its physical meaning. The article provides further discussion about the parameters influenced on fluidity (viscosity for ex.).
Round 2
Reviewer 2 Report
A small reminder:
In the conclusion of the abstract, I recommend inserting (one or two sentences) specific results (numerical values) from the conclusion. Now the conclusion is effective only theoretically.
Thank you for incorporating my comments, I recommend the article for publication.
Author Response
- In the conclusion of the abstract, I recommend inserting (one or two sentences) specific results (numerical values) from the conclusion. Now the conclusion is effective only theoretically.
Answer: The abstract was expanded:
“In the investment casting process, the pattern made of wax is obtained in a die for further formation of shell mold. The problem of die-filling by pattern wax is significant because it influences the quality of the final casting. This work investigates three commercial pattern waxes fluidity with a newly developed injection fluidity test. It was shown that the fluidity of waxes increased with increasing injection temperature and pressure, and the simultaneous increase in temperature and pressure gives a much more significant enhancement of fluidity than an increase in temperature or pressure separately. The rheological behavior of the waxes was also investigated at different temperatures using a rotational viscosimeter, and temperature dependences of waxes dynamic viscosity were determined. It was shown that wax viscosity is increased more than ten times with decreasing temperature from 90 to 60 °C. A good correlation between wax fluidity and its viscosity is observed, which is different from metallic alloys, where the solidification behavior is more critical. The difference in wax flow behavior in comparison with metallic melts is associated with the difference in dynamic viscosity, that for investigated waxes and metallic melts is 3000-27000 mPa·s and 0.5 - 6.5 mPa·s, respectively. The difference in investigated filled waxes fluidity is observed that can be associated with the type and amount of filler. The twice-increasing fraction of cross-linked polystyrene decreases fluidity twice. At the same time, terephthalic acid has a minor influence on wax fluidity.